# The Influence of the Geometry of Movement during the Friction Process on the Change in the Tribological Properties of 30CrNiMo8 Steel in Contact with a G40 Steel Ball

**DOI:** 10.3390/ma17010127

**Published:** 2023-12-26

**Authors:** Marcel Kohutiar, Michal Krbata, Jana Escherova, Maros Eckert, Pavol Mikus, Milan Jus, Miroslav Polášek, Róbert Janík, Andrej Dubec

**Affiliations:** 1Faculty of Special Technology, Alexander Dubcek University of Trencin, Ku Kyselke 469, 911 06 Trencin, Slovakia; michal.krbata@tnuni.sk (M.K.); jana.escherova@tnuni.sk (J.E.); maros.eckert@tnuni.sk (M.E.); pavol.mikus@tnuni.sk (P.M.); milan.jus@tnuni.sk (M.J.); miroslav.polasek@tnuni.sk (M.P.); 2Faculty of Industrial Technologies in Púchov, Alexander Dubček University of Trenčín, Ivana Krasku 491/30, 020 01 Puchov, Slovakia; robert.janik@tnuni.sk (R.J.); andrej.dubec@tnuni.sk (A.D.)

**Keywords:** ball on disc, linear test, steel, wear, coefficient of friction

## Abstract

Experiments with changes in motion geometry can provide valuable data for engineering and development purposes, allowing a better understanding of the influence of tribological factors on the performance and service life of joints. The presented subject article focused on the experimental investigation of the influence of the geometry of the movement of the friction process on the change in the tribological properties of 30CrNiMo8 steel. The friction process was carried out without the use of a lubricant in contact with a steel ball of G40 material with a diameter of 4.76 mm. The steel ball performed two types of movement on the surface of the experimental material. The first method used was ball on disc, in which the ball moved reciprocally in an oval direction at an angle of 180° on a circumferential length of 35 mm at a speed of 5 mm/s. The second method consists of the same input parameters of the measurement, with the difference that the path along which the ball moved had a linear character. The load during the experiment was set at a constant value of 50 N with 1000 repetitions. The results show that with the ball on disc method, there was an increase in wear by 147% compared to the linear test method, which was approximately a coefficient of increase in wear of 2.468. EDS analysis pointed to the occurrence of oxidative wear that affected the resulting COF values, which were lower by 8% when using the ball on disc method due to a more uniform distribution of O and C on the surface of the friction groove where these elements acted as solid microlubricants. With the ball on disc method, defects in the form of microcracks occurred, which affected the reduction in the values of the depth of the affected area of microhardness.

## 1. Introduction

Friction is a term used to describe the force that arises as a result of the movement of two bodies that are in contact and prevents their relative displacement [1]. In practice, it is represented by the coefficient of friction (µ or COF) and is used to express the relationship between the frictional force (tangential force F) and the normal load (N), which acts as a compressive force. The influence of contact materials and their microstructure in the experimental conditions under which the friction process takes place plays the most important role. In the case of tribological testing without the use of lubrication, it occurs between contact metal pairs that are exposed to classic atmospheric conditions to form an oxide layer. These oxides, which occur at the contact interface of friction pairs under the action of lower loads, create a type of protective layer that also acts as a solid lubricant. This statement is also supported by the author G.W. Stachowiak in professional studies [2]. In another experimental study [3], the author D. Arnell agreed with the previous statement of the author [2], and at the same time, in the said study, discussed the concepts of friction and wear arising from tensions at localized contact points between opposing surfaces. The author Ian Hutchings and his team [4] approached the claims of the authors in [2,3] with their research. After increasing the loading force in the friction process, an increase in COF occurred in connection with the breakdown of this thin oxide layer. This finding was also confirmed by expert material in studies [5]. The group of authors led by B. Bhushan in a research study [6] also came to the same conclusion, while the author G. Straffelini came up with the same claim from a scientific study [7].

In the field of materials engineering, surface affinity plays an important role, not only in the design of various functional components, but also in research that affects the friction process. One of the key factors lies in the relationship between the surface geometry and the coefficient of friction, which also affects the wear of the materials themselves. Understanding this relationship is key to optimizing the performance and lifetime of various components. B. Venkataraman et al. [8] provide support for these claims, but also emphasize the importance of understanding this relationship for optimal performance and a longer life of various components. At the same time, the cited scientific study by the collective of authors led by J. Dai [9] in the context of the discussed topic also provides support for their research, which focused on the relationship between surface geometry and the coefficient of friction and supports the aforementioned findings. At the same time, S. Ding and his team [10] also provide support for these claims through their research work, thus emphasizing the importance of the relationship between the surface geometry and the coefficient of friction in the wear process of materials. Most scientists, when investigating tribological properties, focus on different materials or on different changes to the input parameters in experiments. When researching friction properties, they focus mainly on tool steels, which have increased requirements for their service life. The work by Marek Hawryluk et al. [11] focused on the evaluation of the influence of the steel class in terms of tool life in the ball on disc measurement method. The results showed clear traces of abrasive wear and the occurrence of material sticking in these areas. The authors also concluded that a higher tendency to plastic deformation was associated with a lower amount of oxides formed on the steel surface. Another factor is the chromium content in steel, the high value of which can also act against oxidation. However, the chemical composition of the remaining materials showed that this factor is not sufficient to limit the formation of oxides. Similarly, D. Tobola et al. [12] investigated steels with different types and amounts of carbides in contact with Al_2_O_3_ and 100Cr6 balls. The experimental results showed that the sequential turning process could be an alternative to grinding. The differences in wear resistance were closely related to the content and type of carbides. Adhesive wear was the dominant wear mechanism after tests against the 100Cr6 material for all used variants of the mechanical processes. In addition to adhesive wear, two other wear mechanisms were identified: abrasion wear and plastic deformation. Another important parameter entering the friction process from the point of view of wear as well as COF is the thermal machining of materials using cryogenic processing, which has been the focus of many authors including F. Kara et al. [13], B. Wang et al. [14], Ibrahim Gunes et al. [15], and Pello Jimbert et al. [16]. The authors concluded that the deep cryogenic machining process led to the lowest wear and COF values. Fe-19Cr-15Mn-0.66N type steel was investigated by S. Sheng et al. [17]. The results showed that the oxide layers formed by Cr_2_O_3_, Fe_2_O_3_, and Mn_2_O_3_ were favorable for obtaining a low coefficient of friction. Additionally, the effect of sliding speed led to possible changes in COF and wear. This issue was addressed by L. Tang et al. [18], where the authors concluded that the main wear mechanism was adhesive wear in the ball on disc measurement method. In addition, dominant delamination wear occurred. Similarly, M. Ruiz-Andres [19] addressed the issue of the influence of friction speed in the ball on disc method on two-phase DP600 steel. The authors evaluated that the COF and wear rate showed a large dependence on the sliding speed with the occurrence of mainly oxidative wear mechanisms. The investigation of the geometry of the surface or the parameter of the arithmetic mean Ra is a method specified in many scientific works dealing with tribological processes. This leads to the influence of the cutting parameters on the final surface finish. R.B.D. Pereira [20], within the framework of the investigation of the geometry of the surface and the parameter of the arithmetic mean Ra, has made a significant contribution through his scientific work, which focused on the analysis of the influence of the cutting parameters on the final surface treatment. In addition, Todorovic, P. [21] and his scientific colleagues have contributed to this topic through their research works. Their studies deeply investigated the effect of cutting parameters on the surface properties and are widely cited in the context of tribological analyses. The aforementioned scientific study by the collective of authors led by D. García-Jurado [22] dealt with the same issue. All results indicate that the lower the Ra value of the contact surface, the lower the COF and wear value. In his research paper, A. Riyadh [23] specifically supported this correlation between Ra and COF values. His approach to the study of contact surfacing has helped more precisely define the relationship between surface geometry and friction. A scientific study by X. Li [24] also supports these claims. In his research work, X. Li analyzed the influence of Ra values on the tribological properties. His contributions significantly strengthen the argument for a connection between the surface treatment and frictional behavior. The work by P.L. Menezes [25] also examined the results regarding the values of the Ra parameter and their impact on wear. His experimental analysis provides valuable insights into the importance of the optimal surface treatment for minimizing friction and material wear.

Depending on the chosen experimental parameters of the friction pairs, different shapes of the COF curves can arise [26]. In general, however, the COF curve represents the shape shown in Figure 1, which is divided into two main parts. The entire friction process is closely related to wear, which refers to the gradual degradation and loss of material from a solid surface due to repeated contact and friction with another surface or object. Wear can have significant consequences on the performance and lifetime of a component including reducing its efficiency, increasing its energy consumption, and causing structural failure. 

The article in question examines the influence of the geometry of the movement of the friction process on the change in the tribological properties of 30CrNiMo8 steel during dry contact with a steel ball of G40 material. The aim of the presented study was to predict how the direction of the geometry of the movement of the tribological pair affects the resulting tribological properties and also provides a summary view of the mechanics of wear during the given movements. The geometry of the movement, or its current orientation around its own axis, leads to increased wear, which is linked to the emergence of different types of wear. To support these results, COF evaluations as well as EDS analyses of the resulting friction surfaces for selected chemical elements were performed. Finally, measurements of the change in plastic deformation just below the surface of the friction groove were also carried out, which showed different values of the increase in hardness compared to the base material, depending on the type of friction movement. 

Overall, these results can provide valuable information for researchers and engineers interested in the field of tribology and those looking for ways to improve the performance of materials and processes subject to friction with respect to the motion geometry used. Various branches of industry (e.g., automotive, aviation, energy industry) can use these achieved results. When choosing tribological processes for certain applications, the chosen test method may be preferred in order to minimize wear and increase the life of the various components that interact with each other through their functional surfaces. Knowledge of oxidative wear and its effect on COF can lead to the setting of appropriate operating parameters that could reduce friction and wear under similar conditions. Most of the publications that have investigated the geometry of the tribological process in some way have focused more on the investigation of the asperity of the surface in relation to the geometry of the pressure indenter. Regarding the geometry of movement, there is a research gap in this area. Therefore, this study aimed to investigate this problem and determine what differences will arise in the result of wear.

## 2. Materials and Methods

For the experiment, medium-alloyed steel marked 30CrNiMo8 (Q + T) was used. The entire heat treatment process is shown in Figure 2b. Steel is characterized by high values of the fatigue limit in alternating and combined methods of stress (bending, torsion, tension, and shear). It is particularly suitable for heavily stressed parts in aviation, the automotive industry, and the military (e.g., shafts of combat vehicles). The steel also exhibits high strength while maintaining good impact strength. All experiments were carried out at the Faculty of Special Technology in Trenčín, which houses all the measuring devices that were used in this research. The annealed steel was delivered in the form of a rod with a diameter of 55 mm and a length of 500 mm. 

The microstructure of the steel after the heat treatment process is shown in Figure 2b. The material was etched with 3% Nital. The evaluated metallographic image of the experimental steel was in the state after the final refinement of Q + T (Figure 2). This microstructure showed a clearly sorbitic microstructure, where fine globular particles of cementite (Fe_3_C) could be observed, which originated from acicular carbide particles and a ferritic matrix. Moderate prior austenite grain boundaries (PAGB) were also observed. High-temperature tempering was carried out at a temperature of 600 °C (Figure 2a), which allowed good machining to the final dimensions of the samples.

The prescribed chemical composition of the supplied steel was also verified, which is shown in Table 1. The chemical composition was verified using a SPECTROMAXx LMX10 device (SPECTRO Analytical Instruments GmbH, Kleve, Germany). The given table shows the chemical composition according to the DIN standard as well as the supplied material attestation by the manufacturer. Table 2 shows the basic mechanical properties of the material according to the diameter of the rod.

The main experiment was performed on the tribological device UMT TriboLab (Bruker Corporation, Billerica, MA, USA) from Bruker [29,30]. This device offers the possibility of the modular exchange of different parts of the device, which allows for the use of different kinds of tribological process methods. The tribological methods used consisted of a comparison of two measurement methods, namely the ball on disc and the linear test using a hardened steel ball of G40 material with a diameter of 4.76 mm. This material was chosen because it is a common material used in engineering as a bearing material. The measurements took place at room temperature without the use of lubrication. In the ball on disc method, a ball of G40 material moved reciprocally along an oval surface with an angle of 180° on a circumferential length of 35 mm at a speed of 5 mm/s. Due to the prescribed dimensions of the sample, tribological measurements were performed on the same sample that was used on the given device, and therefore a larger measurement angle could not be used in the ball on disc method. The load during the experiment was set at a constant value of 50 N with 1000 repetitions. The load was chosen in order to obtain a sufficient amount of wear and to be able to compare the wear results of both types of measurement. The same damping parameters were also used in the linear test, with the difference that in the given measurement, the ball moved along a linear path. A total of five samples were made for statistical purposes. The samples had a diameter of 50 mm and a thickness of 9 mm. A graphical representation of the friction movements is shown in Figure 3a for a better understanding. The surface of the samples was finely turned after the final heat treatment. The selected sample after measurement is shown in Figure 3b.

An Olympus LEXT OLS5100 confocal microscope (EVIDENT Europe GmbH, Hamburg, Germany) was used to evaluate the wear, with which the 3D topographies of the surface of the base material and the G40 pressure ball were evaluated before the measurements as well as the resulting topographies of the friction grooves.

The chemical composition of the created friction grooves was determined using a thermo-emission scanning electron microscope (TESCAN VEGA 3, TESCAN GROUP, Inc., Kohoutovice, Czech Republic), equipped with an EDS x-act analyzer from Oxford Instruments [31,32]. This analyzer captures the characteristic X-rays emitted from the area that is analyzed using the electron beam of a thermo-emission scanning electron microscope on the surface of the material being examined. Using the method of energy-dispersive spectroscopy (EDS), the obtained X-ray spectrum was analyzed, which was converted into a subsequent monitored and analyzed energy spectrum [33].

## 3. Results and Discussion

### 3.1. Roughness and Hardness of Materials

The topography of the experimental 30CrNiMo8 steel after the finishing turning process is shown in Figure 4a. The value of the resulting roughness, which was achieved by averaging all measured values from five samples, was Sa = 3.34 ± 0.19 µm. Contact pressure balls made of the G40 material showed very low roughness values, namely Sa = 0.22 ± 0.01 µm (Figure 4b). This low value is closely related to the production process of these balls (shaping + grinding + polishing) [34,35,36]. The roughness input values of the G40 steel ball did not have any significant influence on the results, as the material achieved very low Sa values. Therefore, we did not expect any influence on the results of the COF or consumption values. Additionally, a load of up to 50 N was used during the measurement, which is one of the main parameters that affect the entire tribological process. Small COF fluctuations may occur during the initial run-up of the tribological process, but this initial part of the curves will not be taken into the final results.

The hardness of the experimental 30CrNiMo8 steel after high-temperature tempering reached a value of 380 HV10. In the same way, the hardness value of the pressure ball of hardened G40 steel material was measured, which reached an average value of 800 HV10. Since this material showed significantly higher hardness than the experimental material, we can state with certainty that the selected pressure material will act as a friction wedge that will resist wear to a higher degree than the experimental 30CrNiMo8 steel. This is the main condition when designing the measurement of friction pairs, because if the opposite ratio of hardness values occurs, there would be no wear of the investigated experimental material, but significant wear of the pressing friction tool, which was not the subject of this investigation. The G40 material is also used in engineering for the production of steel balls in bearings, therefore it must withstand high loads in the tribological process and achieve significant hardness, which is not least connected to the strength of this material.

### 3.2. Coefficient of Friction and Wear

An overall comparison of the measured coefficients of friction (COF) waveforms obtained after averaging all of the measured results is shown in Figure 5. Two zones can be observed in the figure. Zone A represents the starting part of the curve where a friction groove begins to form on the surface of the material. Zone B represents an already established part of the curve [37]. It was also possible to observe that the maximum value of COF in the ball on disc method was reached after a period of about 1000 s, during which we reached a value of 0.68. Subsequently, the COF value stabilized and fluctuated alternately. This fluctuation in values was caused by the accumulation of microparticles of material in front of the friction ball. This material resists friction. After releasing this material, which takes place in two parts, the first part of the material moves to the edge parts of the friction track, and the second part of the material, which is in a smaller proportion, is pushed into the surface of the friction track by a pressure ball and acts as an abrasive particle. The alternation of these methods of accumulation and the release of microparticles caused the fluctuation of the curve in question [30]. The appropriate COF value for the ball on disc method was 0.59. On the other hand, with the second measurement method, the linear test, the initial increase in COF stopped at the value of 0.62 at a measurement time of 1400 s. With this method, the curve showed a slight gradual increase in the COF value. Its average value in the steady part was 0.64. Here, we can state that with the ball on disc method, we achieved a slight reduction in the COF value compared to the second linear test method. This reduction in COF value was approximately 8%. A prerequisite for a slight decrease in the COF values was probably the occurrence of oxidative wear on the surface of the experimental materials, respectively, on the surface of the G40 pressure ball, which affected the resulting COF values. The rationale for the difference in COF values will be analyzed in the wear mechanisms section. 

An optical comparison of the width of the friction grooves is shown in Figure 6a,b. From the obtained results, it can be clearly observed that the change in the movement of the pressure friction ball led to a significant change in the width and shape of the friction groove. At first glance, it can be clearly observed that with the linear test method, the G40 pressure ball did not reach such a depth, or significantly less wear occurred, than with the ball on disc method. The given images also show red lines that represent the selected 2D groove profiles. A comparison of these two grooves is shown in Figure 6a,b. When comparing these two 2D profiles, it can be seen that with the ball on disc method a maximum groove depth of 108 µm was achieved (Figure 6c), while with the second method, the maximum groove depth was at the level of 71 µm (Figure 6d). We also observed here that peaks, which can occur as a result of the friction process, did not appear on the edges of the friction grooves with either method. This is due to the fact that the material underwent total heat treatment (Q + T) as well as the fact that the speed of movement was not high. An overall comparison of the wear rate after using different measurement methods is shown in Figure 7. With the ball on disc method, the average wear was reached at the level of 1.53 mm^3^, while with the linear test method, the wear was at a value of 0.62 mm^3^. With the ball on disc method, there was an increase in wear by 147% compared to the linear test method.

The analysis of the surface of the selected contact pressure balls is shown in Figure 8a,b. The wear rate was not evaluated on these materials because the balls showed very low wear values that were negligible. Only the topography of the friction surface was evaluated in order to analyze the friction mechanism. As we can see in the first picture, there was a friction surface that was divided into several contact areas. These areas mainly consisted of two friction surfaces where the main wear of the G40 pressure material occurred. This phenomenon is associated with the movement of the friction ball, which was reciprocally rotating at an angle of 180°. On the other hand, with the linear test method, there was one wider area in which there were clear parallel grooves, which were created after the friction process along the reciprocal linear path. 

Figure 9 shows enlarged surfaces of the contact friction of the G40 balls. Since the balls of the G40 material showed much higher hardness than the experimental samples, only shallow grooves were formed on the ball on disc sample (Figure 9a) or deeper parallel grooves in the linear test method (Figure 9b). Oxidative wear accompanied the formation of black surfaces on both surfaces. With the linear test method, it was also possible to observe small pits on the surface, which are characteristic of the rolling of abrasive particles during three body abrasion wear [38,39,40].

### 3.3. Wear Mechanisms

The surface of two selected friction surfaces was examined using SEM. Figure 10a, in which the friction process was measured using the ball on disc method, showed signs of delamination in the form of small torn shells in addition to shallower parallel grooves. These thin shells can also be seen in the given figure, which later in the process support the formation of abrasive particles or abrasive wear [41]. During this process, microcracks also appear on the friction surface, which arise due to higher plastic deformation of the surface. These microcracks usually occur in perpendicular directions with respect to the focus of the friction mechanism, and when they are cut, thin parts of the material that resemble a shell can tear off. These thin shells of material can subsequently be gradually crushed, transforming them into hard abrasive microparticles [42,43]. With this method, the surface showed the resulting combined wear, a combination of adhesive and abrasive. Oxidative wear was observed, but to a much lower extent than with the linear test method (Figure 10b). With this method, oxidative wear occurred in larger areas, which is shown in a darker color in the figure. The surface of the friction groove after the linear test also showed higher deep parallel abrasive grooves than during the ball on disc test. Since in the linear test there is a reciprocal movement along the same track, these grooves showed a much greater depth. Additionally, as in the previous case, loose debris was found on the surface. When comparing both surfaces, we noted a clearly different surface morphology, which in the case of the linear test sample consisted of large islands alternating with hollowed-out valleys. The resulting wear on the given surface is mainly represented by abrasive wear, which was manifested by the formation of deep grooves in combination with more pronounced areas of oxidative wear.

Comparison of the EDS maps of selected elements in friction grooves is shown in Figure 11. The distribution of Fe (Figure 11a,b) was uniform, in line with the fact that in the ball on disc method, there were empty small dark islands on the map in which compounds based on Si + O were formed. These empty islands were also found in the linear test method but in smaller fractions. With this method, we also observed the occurrence of extensive dark areas where Fe chemically combined and formed compounds with O (Figure 11b). This means that with both methods, an oxidation layer appeared on the surface, with the difference that in the ball on disc method, O was more regularly distributed on the surface (Figure 11c). In the case of the linear method, the occurrence of O was distributed into smaller concentrated areas (Figure 11d). The main alloying elements Cr, Ni, Mo, and Mn were evenly distributed on the surface in both cases, and the process did not have a significant effect on the change in their distribution (Figure 11e–l). As already mentioned, the occurrence of Si was concentrated in small globular areas (Figure 11m,n), which represent small detached abrasive particles, which participated in the further process of increasing the wear of friction pairs [44,45]. The occurrence of C on the surface of the friction surfaces is shown in Figure 11o,p. With the ball on disc method (Figure 11o). the carbon was distributed more evenly on the surface than with the linear test method (Figure 11p). Carbon is used as a solid lubricant mainly for components that are exposed to elevated temperatures [46,47,48]. Therefore, it can be concluded that the reduction in COF values in favor of the ball on disc method is caused by a larger and more uniform distribution of this element on the surface of the friction surface.

### 3.4. Microhardness of the Surface after Wear

After wear, the microhardness distribution just in the subsurface layer was measured using the worn samples. Figure 12 shows the microhardness of the subsurface of the contact surface, which was perpendicular to the direction of the friction surface. Each point represents the mean value of five measurements taken at different locations with the same distance from the surface. The microhardness value of the base material reached an average value of 383 HV0.05. With the ball on disc method, the highest hardness of 428 HV0.05 was achieved at a depth of 0.01 mm. Subsequently, the hardness continuously decreased to the basic hardness value. The depth of the hardened layer reached a value of 0.19 mm. With the second linear test method, a higher hardness value was reached just below the surface of the friction groove, up to 510 HV0.05 at the same depth. The hardness again decreased continuously with the difference that the affected depth under the friction groove reached a value of 0.28 mm. The change in hardness under the friction grooves is due to two reasons. One is the very effect of the movement of the friction pair, while the other is the deformation strengthening due to strong surface plastic deformation [49]. 

This contrasting result is due to a combination of several results. One of them is the formation of undesirable microstructural defects such as microcracks and porosity due to the intensity of plastic deformation. As is known, the rate of plastic deformation is closely related to the kinetic energy [50] as a function of the size of the pressure particles and the speed of the movement of the friction materials [51,52]. 

Another is the different rate of formation of chemical compounds that can occur just below the surface of the grooves that are affected by the direction of movement of the friction pair, which last but not least, affect the deformation behavior of the material.

## 4. Conclusions

The article in question examined the influence of the geometry of the movement of the friction process on the change in the tribological properties of 30CrNiMo8 steel during dry contact with a steel ball composed of G40 material. The aim of the presented study was to predict how the direction of the geometry of the movement of the tribological pair affected the resulting tribological properties. The evaluation also focused on the comparison of COF, the wear mechanics, and EDS analysis of the resulting friction surfaces for selected chemical elements as well as the measurement of the change in plastic deformation using the microhardness measurement just below the surface of the friction groove. The following conclusions can be drawn from the presented work:The pressure G40 steel ball is an ideal material for wear testing because it resists wear more effectively compared to the experimental 30CrNiMo8 steel due to its ability to act as an effective friction wedge;In the ball on disc method, the material achieved an 8% lower friction coefficient compared to the linear test due to a more uniform distribution of O and C on the surface of the friction groove, while these elements function as solid microlubricants;Changing the movement of the pressure friction ball led to a significant change in the width or shape of the friction groove. The ball on disc method achieved a higher maximum friction groove depth (108 µm) compared to the linear test method (71 µm), which significantly affected the overall wear of the material;With the ball on disc method, an average wear of 1.53 mm³ was achieved, which represents an increase of 147% compared to the linear test (0.62 mm³). The coefficient of increase in op-ordering was approximately 2.468;The hollowed-out grooves of the G40 pressure balls showed shallow grooves in the ball on disc method and deeper parallel grooves in the linear test. Both surfaces showed the presence of oxidative wear. In the linear test, small pits were observed, characteristic of the rolling of abrasive particles during three-body wear;Combined wear including adhesive and abrasive wear was observed in the ball on disc method. The process was accompanied by the formation of microcracks due to higher plastic deformation of the friction surface. In contrast, in the linear test method, mainly abrasive wear prevailed with the formation of deep grooves and significant areas of oxidative wear;Oxidative wear was observed with both test methods. With the ball on disc method, this wear was less common than with the linear test. In this method, oxidative wear occurred in larger concentrated areas, as evidenced by the EDS analysis;Even distribution of the main alloying elements on the surface (Cr, Ni, Mo, Mn) did not significantly affect the friction process. The distribution of Fe was uniform, with the exception of small spherical islands in the ball on disc method, where compounds based on Si + O were formed. These islands, representing small abrasive particles, can increase the wear of friction pairs in the further process, which was also observed in the linear test methods, but to a lesser extent;Microhardness results showed that the linear test method caused a greater depth of the formed subsurface layer (0.28 mm) due to plastic deformation of the material compared to the ball on disc method, where the depth of the affected area was only 0.19 mm. In the ball on disc method, defects were created in the form of microcracks, which resulted in a decrease in the microhardness values.

It can be concluded that changing the geometric direction of the friction path while maintaining all of the measurement parameters in the friction process leads to different COF values as well as the resulting wear. In the future, it would be appropriate to investigate this experimentally.

## Figures and Tables

**Figure 1 materials-17-00127-f001:**
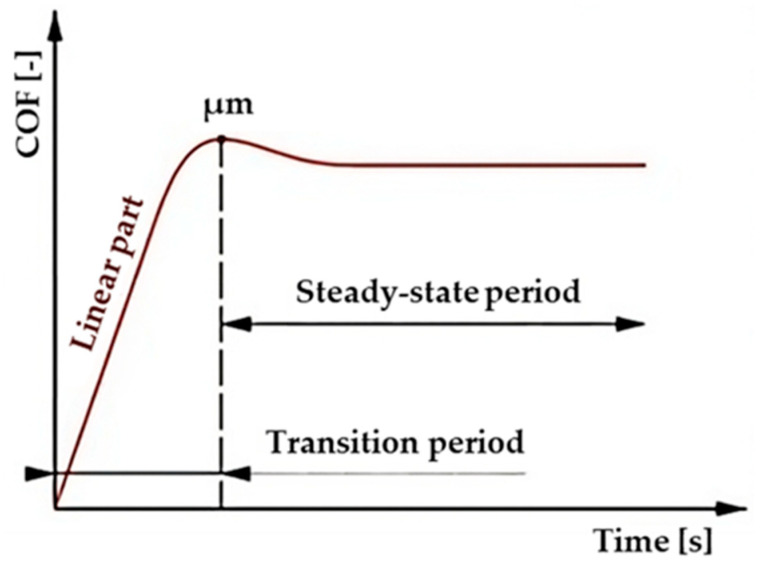
Typical curve of coefficient of friction vs. time [27,28].

**Figure 2 materials-17-00127-f002:**
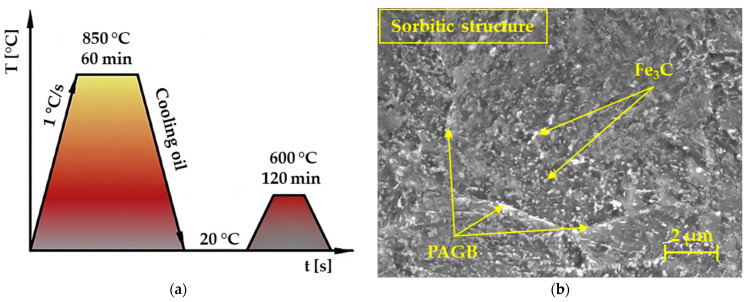
(**a**) Schematic diagram of the heat treatment of 30CrNiMo8 steel. (**b**) Microstructure of 30CrNiMo8 steel after final heat treatment.

**Figure 3 materials-17-00127-f003:**
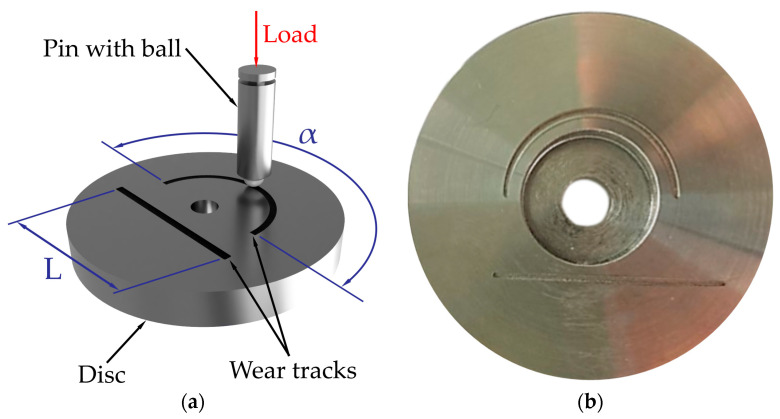
(**a**) A graphical representation of the friction movements. (**b**) Sample of the 30CrNiMo8 material after tribological measurements.

**Figure 4 materials-17-00127-f004:**
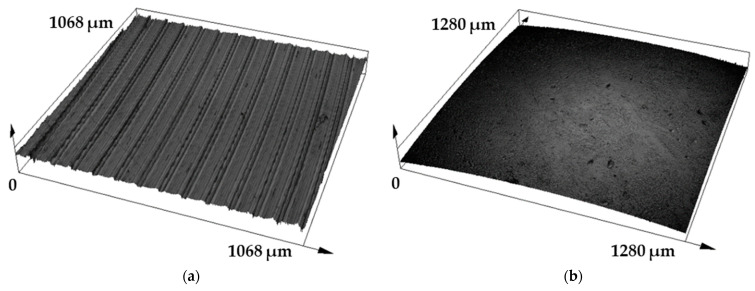
Surface texture evaluated using a confocal microscope: (**a**) 30CrNiMo8—Sa = 3.34 µm; (**b**) G40 steel—Sa = 0.22 µm.

**Figure 5 materials-17-00127-f005:**
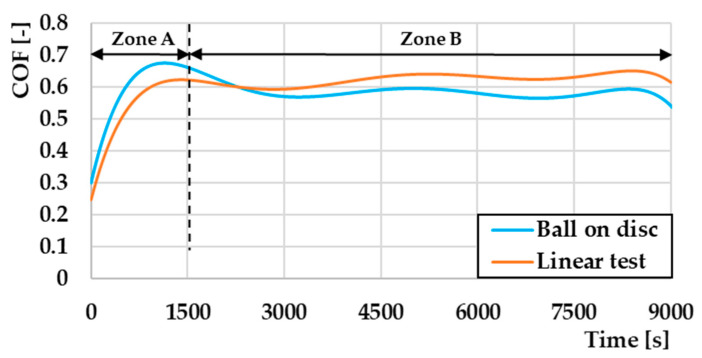
Comparison of COF values depending on the measurement methodology.

**Figure 6 materials-17-00127-f006:**
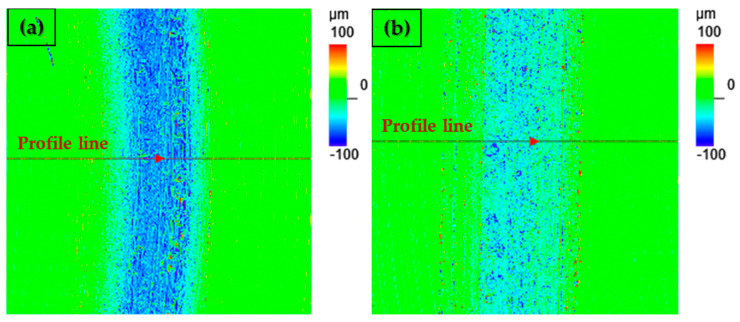
Comparison of friction grooves. (**a**) Height morphology of friction grooves for the ball on disc test; (**b**) height morphology of friction grooves for the linear test; (**c**) 2D profile of the groove created during ball on disc; (**d**) 2D profile of the groove created during the linear test.

**Figure 7 materials-17-00127-f007:**
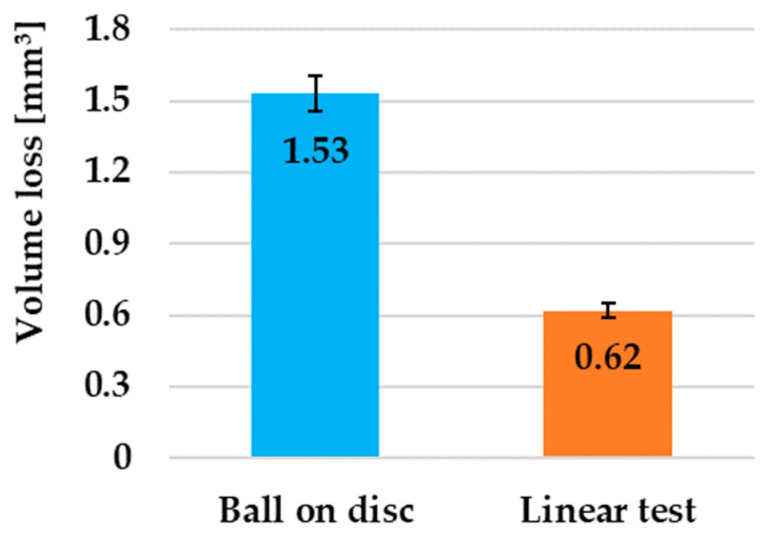
Overall comparison of the wear rates after using different measurement methods.

**Figure 8 materials-17-00127-f008:**
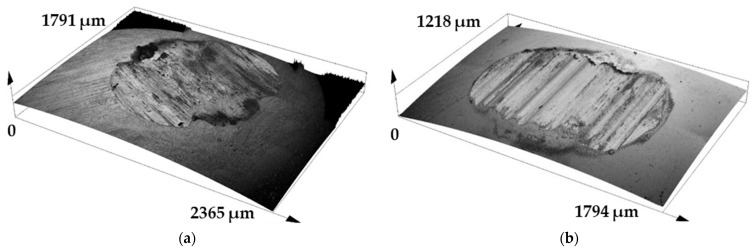
Ball wear after test: (**a**) ball on disc; (**b**) linear test.

**Figure 9 materials-17-00127-f009:**
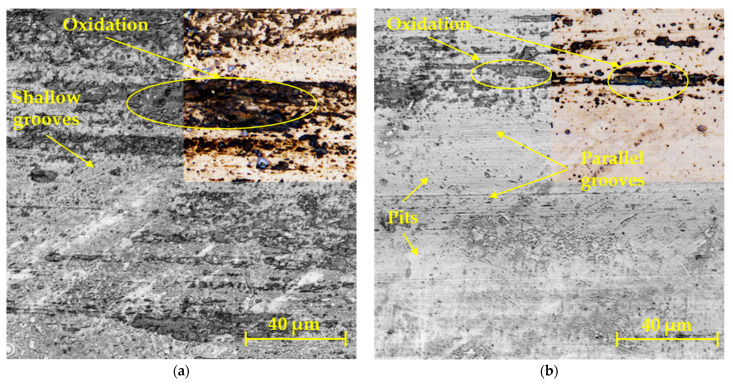
Worn surfaces of the G40 balls in methods: (**a**) ball on disc; (**b**) linear test.

**Figure 10 materials-17-00127-f010:**
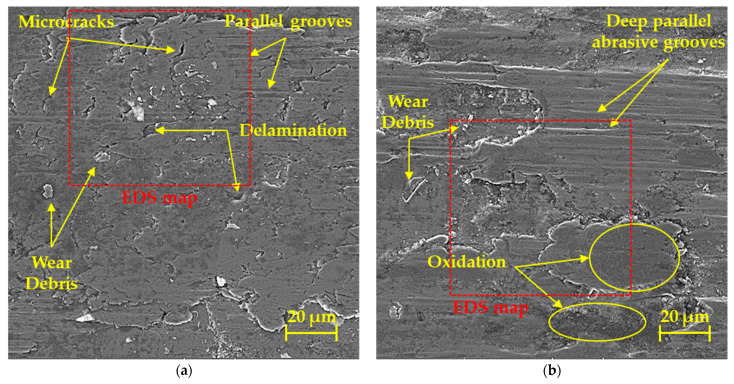
Worn surfaces of friction grooves: (**a**) ball on disc; (**b**) linear test.

**Figure 11 materials-17-00127-f011:**
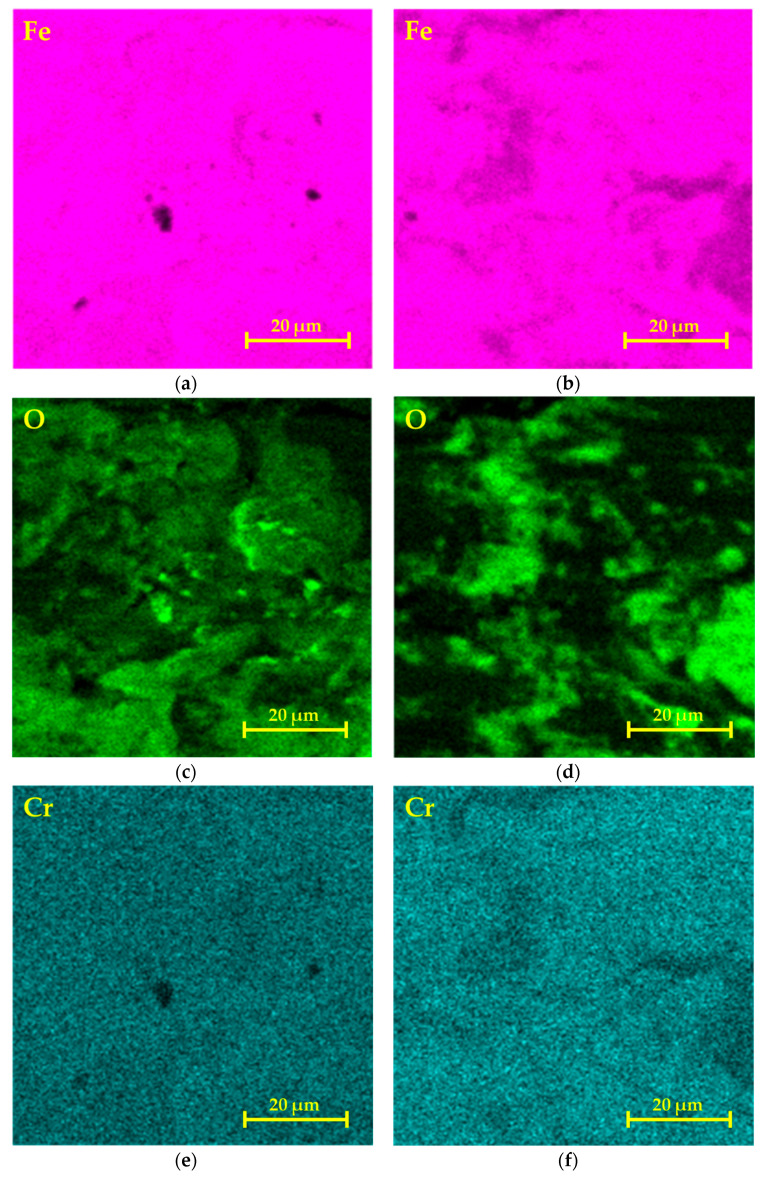
Comparison of EDS maps of the distribution of selected chemical elements from friction areas: ball on disc (**a**,**c**,**e**,**g**,**i**,**k**,**m**,**o**); linear test (**b**,**d**,**f**,**h**,**j**,**l**,**n**,**p**).

**Figure 12 materials-17-00127-f012:**
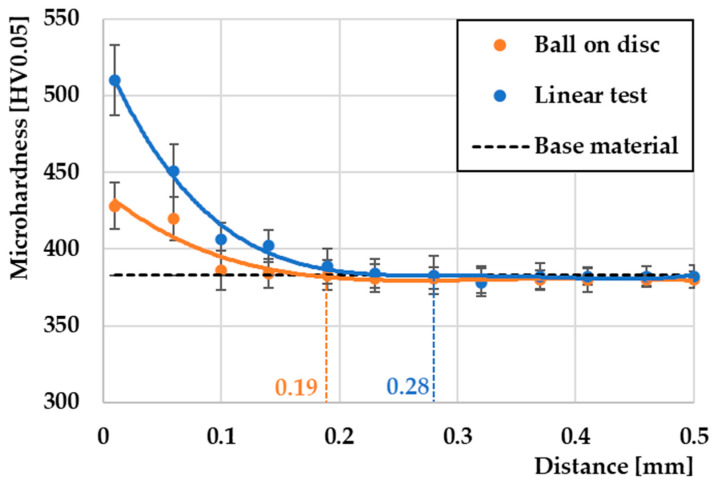
Comparison of microhardness of plastically deformed surfaces of friction grooves.

**Table 1 materials-17-00127-t001:** Composition of 30CrNiMo8 steel, wt.%.

30CrNiMo8	C	Mn	P	S	Si	Ni	Cr	Mo
Standardized material composition [%]	0.26–0.34	0.3–0.6	max. 0.035	max. 0.035	0–0.40	1.8–2.2	1.8–2.2	0.3–0.5
Material certificate [%]	0.32	0.54	0.016	0.026	0.22	1.98	2.11	0.377
Spectral analysis [%]	0.34	0.52	0.002	0.024	0.21	1.86	2.08	0.42

**Table 2 materials-17-00127-t002:** Parameters of 30CrNiMo8 steel.

Average [mm]	To 16	16–40	40–100	100–160	160–250
Rm [MPa] (Q + T)	1250–1450	1250–1450	1000–1300	1000–1200	850–1100
Re [MPa] (Q + T)	1050	1050	900	800	700
KV [J] (Q + T) + 20 °C	30–45				
A [min. %] (Q + T)	8–14				
Z [%] (Q + T)	40	40	45	50	50

## Data Availability

Data are contained within the article.

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
