# Peer review of "The Influence of the Geometry of Movement during the Friction Process on the Change in the Tribological Properties of 30CrNiMo8 Steel in Contact with a G40 Steel Ball"

_materials, 2023, doi:10.3390/ma17010127_

Round 1

Reviewer 1 Report

Comments and Suggestions for Authors

materials-2762701

In the manuscript titled “The Influence of the Geometry of Movement During the Friction Process on the Change in the Tribological Properties of 30CrNiMo8 Steel in Contact with a G40 Steel Ball”, the authors investigate how the geometry of the movement affects the tribological properties of the 30CrNiMo8 steel during dry sliding friction with a steel ball. They compare the results of two tribological methods: Ball on disc and Linear test.

This topic is important and relevant for the field of tribological characterization of 30CrNiMo8 steel; however, the manuscript also has some limitations and weaknesses that need to be addressed and improved. Here are some specific recommendations and suggestions for each section of the manuscript:

Introduction

1.       The authors of this article need to give a clear research gap in their introduction section to justify the significance and purpose of their study.

2.       The authors need to specify what are the main objectives and contributions of their article at the end of the Introduction section, by stating clearly what they aim to accomplish and what they expect prove in their research.

Materials and Methods

3.       The authors need to present the criteria for selecting the test parameters by citing relevant sources.

4.       Since the surface roughness is an important factor that affects the tribological properties of the materials, as the authors mentioned in their introduction section it is necessary to include in this section the surface roughness of the two bodies before conducting the wear experiments.

The manuscript needs major revision before it can be considered for publication. The authors must address all the recommendations and suggestions mentioned above to improve their manuscript.

Reviewer 2 Report

Comments and Suggestions for Authors

This article is comprehensive, logically organized, and contains valuable information on the influence of the geometry of movement during the friction process on the change in the tribological properties of 30CrNiMo8 steel in contact with a G40 steel ball.

To improve the manuscript, the authors should take the following considerations:

(1) The authors presented the worn surfaces of friction grooves: (a) Ball on disc; (b) Linear test using a SEM microscope in Figure 10. The authors should provide and discuss worn surfaces of friction grooves: Ball on disc; and Linear test using a transmission electron microscopy (TEM) microscope as well to gain a better understanding of the microstructure of surfaces.

(2) The authors presented the comparison of microhardness of plastically deformed surfaces of friction grooves in Figure 12. However, this manuscript does not contain much error analysis on the composite performance which is highly required for readability purposes. It is suggested the authors should place the standard deviations of the microhardness of plastically deformed surfaces of friction grooves for the reliability and readability of the present research.

The submitted manuscript has significant scientific insights and the conclusions are soundly supported by the experimental data. However, the manuscript requires minor revisions before being accepted in the Special Issue: Research on Tribology and Anti-wear Behavior of Metals and Alloys in the well-circulated journal, Materials.

Reviewer 3 Report

Comments and Suggestions for Authors

Dear Author(s), the manuscript ‘The Influence of the Geometry of Movement During the Friction Process on the Change in the Tribological Properties of 30CrNiMo8 Steel in Contact with a G40 Steel Ball’, Manuscript ID: materials-2762701, have some weakness that must be revised appropiately.

Please find below some, of the most crucial comments:

1.      In the Abstract section, Authors shuld introduce firstly to the area of research and presenting the significance of the study. Currently, there is no goal put in the paper as the main line.

2.      Secondly, still to the Abstract section, the significance of the analysis must be highlighted. Authors did not present the industry or practical applications to the advantegous of the results obtained.

3.      For the Introduction section, each of the reffered items must be cited separately, like [2-4] or [5-7]. Authors should present advantages and disadvantages of the previos study, which can indicate the meaning of the proposal.

4.      The motivation described in lines 97-108 does not derive from the critical review of the previously published papers. Usually, in the scientific paper, the novelty is suggested according to the lack in the current state of knowledge. From that meaning, the proposal is required. Authots did not clearly define the disparencies from the previous references.

5.      Some values of sets of experiment, described in lines 138-166, are not fully justified and looks like selected arbitrarily. Please add some irrelevant information with judgements completed.

6.      Along with the section 2 . Materials and Methods, an accuracy of the Olympus LEXT OLS5100 confocal microscope measurement was not recognized. What about the uncertainty or noise or other type of errors? Please try to refer to the measurement process more comprehensively, e.g.:

(1)   https://www.doi.org/10.1088/2051-672X/3/3/035004

(2)   https://www.doi.org/10.1016/j.measurement.2023.113853

(3)   https://www.doi.org/10.1007/s41871-020-00057-4

7.      Considering the subsection 3.1 Roughness and Hardness of materials, why only a Sa roughness parameter was studied? Justification of the only selected ISO 25178 roughness indicators against others must be risen.

8.      The critical discussion in subsection 3.2 Coefficient of friction and wear, is not provided. In that meaning, the advantaegs and any potential disadvantages of the study is unknown. Author have more 3rd section on the results presentation than any discussion.

9.      Each of the advantages and disadvantages must be justified and discussed more clearly, especially in subsections 3.2, 3.3 and 3.4 – it does not exist. The advantages must be highlighted in the meaning of both mentioned issues.

10.  The 5. Conclusions section is long. All of the informations provided can be presented, however Authors must emphasize the novelty and main purpose against previous studies. In this section both general and detailed information must be eparated that the reader would not lose what the Authors are trying to convey. This section requires strong improvement.

From the above, the reviewed manuscript must be improved significantly before any further processing, if allowed by the Editor.

Round 2

Reviewer 1 Report

Comments and Suggestions for Authors

Accept in present form

Reviewer 3 Report

Comments and Suggestions for Authors

Dear Authors,

The manuscript was improved in a proper manner.

All of the raised comments were responded appropiately resolving the issues listed so, respectively, the submission can be considered for publication in its current, revised form.

Regards,

Reviewer